# SARS-CoV-2 sequencing artifacts associated with targeted PCR enrichment and read mapping

Kirsten Maren Ellegaard[1]*, Vithiagaran Gunalan[2], Raphael Sieber[1], Sharmin Jamshid Baig[1], Nicolai Balle Larsen[2], Marc Bennedbæk[2], Jonas Bybjerg-Grauholm[3], Leandro Andrés Escobar-Herrera[1], Tobias Gress[2], Theis Hass Thorsen[1], Anders Krusager[2], Gitte Nygaard Aasbjerg[1], Nour Saad Al-Tamimi[3], Casper Westergaard[1], Christina Wiid Svarrer[4], Morten Rasmussen[2], Marc Stegger[1,5]

1 Department of Sequencing and Bioinformatics, Statens Serum Institut, Copenhagen, Denmark, 2 Department of Virology and Microbiological Preparedness, Statens Serum Institut, Copenhagen, Denmark, 3 Department of Congenital Disorders, Statens Serum Institut, Copenhagen, Denmark, 4 Department 318 CMC Bioanalysis, Novo Nordisk, Copenhagen, Denmark, 5 Antimicrobial Resistance and Infectious Diseases Laboratory, Harry Butler Institute, Murdoch University, Perth, Australia

* krel@ssi.dk

## Abstract

Protocols and pipelines for SARS-CoV-2 genome sequencing were rapidly established when the COVID-19 outbreak was declared a pandemic. The most widely used approach for sequencing SARS-CoV-2 includes targeted enrichment by PCR, followed by shotgun sequencing and reference-based genome assembly. As the continued surveillance of SARS-CoV-2 worldwide is transitioning towards a lower level of intensity, it is timely to re-visit the sequencing protocols and pipelines established during the acute phase of the pandemic. In the current study, we have investigated the impact of primer scheme and reference genome choice by sequencing samples with multiple primer schemes (Artic V3, V4.1 and V5.3.2) and re-processing reads with multiple reference genomes. We have also analysed the temporal development in ambiguous base calls during the emergence of the BA.2.86.x variant. We found that the primers used for targeted enrichment can result in recurrent ambiguous base calls, which can accumulate rapidly in response to the emergence of a new variant. We also found examples of consistent base calling errors, associated with PCR artifacts and amplicon drop-out. Similarly, misalignments and partially mapped reads on the reference genome resulted in ambiguous base calls, as well as defining mutations being omitted from the assembly. These findings highlight some key limitations of using targeted enrichment by PCR and reference-based genome assembly for sequencing SARS-CoV-2, and the importance of continuously monitoring and updating primer schemes and bioinformatic pipelines.

**Data availability statement:** The data underlying the results presented in the study have been deposited in the European Nucleotide Archive (ENA) at EMBL-EBI under accession number PRJEB78355 (https://www.ebi.ac.uk/ena/browser/view/PRJEB78355).

**Funding:** The author(s) received no specific funding for this work.

**Competing interests:** The authors have declared that no competing interests exist.

## Introduction

The declaration of the COVID-19 outbreak as a pandemic in March 2020 by the WHO [1] resulted in the rapid establishment of protocols and pipelines for genome sequencing of SARS-CoV-2. Five years later, SARS-CoV-2 has evolved to be more transmissible but also less severe, in part due to increased population immunity [2], and in response the global sequencing effort has been scaled down. Still, given that the evolutionary trajectory of the virus remains unpredictable [3], continued surveillance is essential, even if at a lower level of intensity. It is therefore timely to re-visit genome sequencing protocols and pipelines established during the acute phase of the pandemic.

Targeted enrichment by polymerase chain reaction (PCR) is widely used when sequencing viral genomes [4,5], and is by far the most common strategy for sequencing SARS-CoV-2 genomes to date. This sequencing approach has the advantage of being both cheaper and faster than other available methods, as well as enabling sequencing from samples with low viral load [5]. However, the continuous evolution of SARS-CoV-2 is a challenge for targeted amplification, due to mutations coinciding with primer binding sites. If such mutations occur towards the 3'-end of a primer, the PCR reaction of the corresponding amplicon is likely to perform poorly, in which case a "spike-in" primer or a new primer scheme will be required. The most widely used primer scheme for SARS-CoV-2, developed by the Artic Network [6], has been updated several times during the pandemic, but despite the enormous effort it has proven difficult to keep up with viral evolution [7]. Not all mutations occurring in primer regions will have a notable impact on PCR amplification, in which case no action is required. However, for such mutations, the PCR enriched sample will contain DNA with bases distinct from the target viral sequence (originating from the primers), which must be removed during processing, as it can otherwise result in genome assembly errors [8].

Targeted enrichment by PCR can also introduce artifacts due to amplification errors [9]. The approach consists of multiple overlapping amplicons, which are multiplexed in two separate pools to avoid "template-switching" and other interactions between primers [5,9]. Still, even partial sequence similarity between the primers can potentially generate unintended PCR products. PCR amplification also occurs during library preparation (when using tagmentation-based kits), at which point the two pools are combined, leaving ample opportunity for chimera formation [10].

The evolution of SARS-CoV-2 also impacts bioinformatic pipelines used for assembling the reads. The most common strategy for assembling SARS-CoV-2 genomes is to use a reference genome for read mapping. This method has the advantage of being robust to variation in coverage, with base calling being possible with as little as 10x coverage. Thus, a complete genome can often be obtained even with some weakly-performing amplicons. Moreover, a single continuous sequence (contig) will be generated regardless of gaps in genome coverage, greatly facilitating downstream analysis. Since the gene content of viruses like SARS-CoV-2 is very stable, evolutionary change is expected to be mostly point mutations and smaller insertions/deletions, which can, in principle, be captured with a reference-based assembly.

However, in practice, the quality of a reference-based genome assembly depends on the correct mapping of the reads, which will become more challenging as the genetic distance between reads and reference genome increases [11].

At the Statens Serum Institut (SSI) in Denmark, routine SARS-CoV-2 genome-based surveillance has been done on positive PCR-test samples since June 2021, on Illumina platforms using the Artic V3 and V5.3.2 primer schemes. To keep up with SARS-CoV-2 evolution, protocols and bioinformatic pipelines have been continuously updated. For example, custom "spike-in" primers were developed and implemented rapidly in response to emerging variants (with major updates required for the Omicron variant) (S1 Table), and the Wuhan-Hu-1 reference genome was replaced with an Omicron (BA2) consensus genome in October 2023.

In the current study, we investigated the impact of targeted PCR enrichment and choice of reference genome, by sequencing samples with multiple primer schemes and re-processing samples with multiple reference genomes. Additionally, we analysed the temporal development in ambiguous base calls in response to the emergence of the BA.2.86.x variant in Denmark.

## Methods

### Ethics statement

This article has been prepared on the basis of a study carried out as part of a task imposed on the Statens Serum Institut according to national legislation. Therefore, according to the Danish Health care act (§ 222), no approval requirements from the ethics committees is obliged. Most of the authors have access to a database that connects sample-ids to individual participants, due to their involvement in routine surveillance tasks at SSI. The database was accessed on 22nd March and 28th of June 2024, to pull information on genome sequences, including sampling date, Nextclade results and quality metrics. No personal information was extracted, as it had no relevance for the current study.

### Experimental design

Three datasets were analysed in the current study, as outlined in Table 1. First, to investigate whether samples sequenced with different primer schemes generate identical genomes, a total of 810 unique samples were analysed, where each sample was sequenced with two different primer schemes from the same RNA extraction. Second, to test the impact of reference genome choice, 14 samples of the BA.2.86.x variant sequenced using the Artic V5.3.2 primer scheme were processed with three different reference genomes. Third, to investigate whether ambiguous base calls accumulate over time in response to the emergence of a new variant, 8865 samples sequenced as part of the routine SARS-CoV-2 genome surveillance were analysed for ambiguous base calls.

**Table 1. Analysis overview.**

| Analysis | Sampling date | Primer-scheme[a] | Reference genome[b] | No. of samples | Variant | Ambiguity threshold |
|---|---|---|---|---|---|---|
| Primer scheme | 18 Dec 2022 to 26 Jan 2023 | Artic V3 with spike-ins, Artic V4.1 | Wuhan-Hu-1 (NC_045512.2) | 580 | Various Omicron (before BA.2.86.x) | 0.9 |
| | 25 Sep 2022 to 20 May 2023 | Artic V3 with spike-ins, Artic V5.3.2 | Wuhan-Hu-1 | 230 | | |
| Reference genome | 19 Dec 2022 to 20 Dec 2022 | Artic V5.3.2 | Wuhan-Hu-1, BA2 consensus, JN.1.4 | 14 | BA.2.86.x | 0.9 |
| Ambiguous base calls over time | 12 Sep 2023 to 12 Feb 2024 | Artic V5.3.2 | BA2 consensus | 8865 | Various Omicron (including BA.2.86.x) | 0.8 |

[a.] Spike-in primers for Artic V3 are detailed in S1 Table.

[b.] See methods section "Preparation of alternative reference genomes" for details.

## Sample preparation and sequencing

All samples analysed in the current study were collected as part of the national Danish screening efforts during the SARS-CoV-2 pandemic in either the "healthcare track" at Danish hospitals or in the "community track". In both tracks, nasal swabs were stored in sterile tubes and sent to SSI for diagnostic analysis and subsequent sequencing [12].

Upon arrival at TestCenter Denmark (TCDK), samples were registered and processed in a semi-automated laboratory flow. To release viral particles from the swabs, 700 µL of Dulbeco's phosphate-buffered saline (DPBS) buffer were added to individual sample tubes, and the mixture was agitated at 700 RPM for 10 minutes on Hamilton STAR liquid handlers (Hamilton Bonaduz, Switzerland). Samples were analysed for SARS-CoV-2 as described earlier [13]. For each SARS-CoV-2 positive sample selected for sequencing, 200 µL of the mixture was transferred to 96 well plates. For each plate, four negative controls (containing DPBS) were included in wells C04, C09, F04 and F09. Additionally, two negative controls were added in wells G12 and H12, introduced at the time of targeted PCR enrichment and library preparation, respectively.

Total nucleic acids were extracted and purified on the Biomek i7 (Beckman Coulter, Brea, CA, USA)) platform with the RNAdvance Blood kit (Beckman Coulter), and eluted in 50 µL DNase- and RNase-free water. An aliquot of the purified extract was used for variant analysis via an allelic discrimination RT-qPCR assay [14,15], which also functioned as a contamination check and a proxy of sample concentration. cDNA synthesis and SARS-CoV-2 targeted PCR enrichment was done with Illumina's COVIDSeq Test kit, Ruo version (REF#: 20043675, Illumina, San Diego, CA, USA) on Hamilton VANTAGE platforms with an adjusted protocol accommodating for automated sample handling on liquid handlers, otherwise according to manufacturer's instructions. In brief, cDNA was synthesized from 9 µL total nucleic acids per sample. Spike-in primers for primer scheme Artic V3 (S1 Table) were ordered from TAG Copenhagen A/S (TAG Copenhagen A/S, Copenhagen, Denmark) as solid custom 0,20 µmol HPLC-purified DNA oligos, resuspended at 100 µM in DNase- and RNase-free water and diluted to 10 µM to create working stocks. Spike-in primers were added to the master mixes to the concentrations specified in S1 Table. Artic V4.1 and V5.3.2 primer pools were ordered as multiplex PCR panels from Integrated DNA Technologies (V4.1 REF#: 10011442, V5.3.2 REF#: 10016495, Integrated DNA Technologies, Inc., Coralville, IA, USA) with each panel provided individually as two pools at 100 µM total oligo concentration in IDTE buffer, respectively. Both panel versions were diluted to 10 µM to create working stocks. Targeted PCR enrichment was carried out in two separate PCR reaction pools per sample, using 5 µL cDNA extract and 20 µL master mix for each primer pool. The polymerase was activated at 98°C for 3 minutes and amplicons generated by 35 rounds of thermal cycling at 95°C for 15 seconds and 63°C for 5 minutes. Following targeted PCR enrichment, 20 µL from corresponding sample wells were combined to reach a total of 40 µL amplicon extract per sample.

Library preparation was also done with Illumina's COVIDSeq Test kit, but using IDT Index PCR Dual sets (Illumina, San Diego, California, USA) for a subset of the samples to enable higher levels of multiplexing. Sequencing was performed with 2 x 74 bp paired-end reads on a NextSeq 550 sequencing system using a NextSeq 500/550 v2.5 150-cycle Mid Output kit (Illumina).

## Preparation of alternative reference genomes

Two alternatives to the SARS-CoV-2 reference strain Wuhan-Hu-1 (GenBank accession no. NC_045512.2) were analysed in the current study: a consensus genome of BA.2.x genomes (BA2 consensus), and a representative genome of the BA.2.86.x variant (JN.1.4, sampled 27/2–2024). To generate the consensus genome, a collection of 75 high-quality genomes of the BA.2 variant was downloaded from GISAID in September 2023 (10.55876/gis8.240517ku, also see S1 File). A consensus genome was generated from this collection, together with the Wuhan-Hu-1 reference genome (to fill in the ends of the genome not covered by PCR primers). Undetermined bases (Ns) were manually removed from the reference. Additionally, a 9 bp deletion at position 21,633−21,641 was removed as this deletion appeared to be fixed. Similarly, for the JN.1.4 genome, undetermined positions were manually removed, and the sequence was patched at the ends using

Wuhan-Hu-1. Since the alternative reference genomes contain deletions relative to Wuhan-Hu-1, new BED files were constructed to adjust for the changed primer positions (to ensure correct primer trimming on BAM files).

## Genome assembly pipeline

All samples were processed according to the current pipeline for routine SARS-CoV-2 genome surveillance at SSI, except for the ambiguity threshold, which was set to either 0.8 or 0.9 depending on the analysis (Table 1). All commands used are detailed in S2 File.

Initial quality trimming was done with Trim Galore (version 0.6.4, RRID: SCR_011847) [16], using default settings for paired-end reads. Trimmed reads were mapped to the reference genome, using BWA-MEM (version 0.7.17-r1188, RRID: SCR_010910) [17] with default settings. Primer trimming was done with iVar trim (version 1.3.1, RRID: SCR_024045) [8] using the "-e" flag to keep reads without primer sequences, a phred score quality threshold of 20 and a minimum read-length after trimming of 30 bp. SAMtools (version 1.14, RRID: SCR_002105) [18] was used to sort and index BAM files, remove unmapped reads, and count mapped reads.

Consensus genomes were generated from the primer trimmed BAM files using bcftools (version 1.14) [19]. First, a VCF file was generated, using a series of piped bcftools commands (S2 File). The plugin bcftools +setGT was used to over-write the default GT value (genotype), so that the GT value was set to "1/1" for variant allele frequencies equal to or higher than the ambiguity threshold, and to "0/1" when lower than the ambiguity threshold. Indels (insertions and deletions) were accepted based on a minimum IMF value ("maximum fraction of reads supporting an indel") of 0.5, with candidate indels falling below this score being filtered off the VCF file. From the VCF file, the genome sequence was generated with bcftools consensus, with a minimum depth threshold of 10. Ambiguous base calls were specified using the IUPAC ambiguity code, whenever a variant in the VCF file had the GT-value "0/1".

Following genome assembly, all genomes went through a quality check where they were assigned "pass" or "fail" status. Genomes with more than 3000 undetermined bases ("N") were assigned "fail". Samples were further filtered based on the number of mapped reads in samples and negative controls. For each 96-well plate, the $25^{th}$ percentile (q1) and lower whisker point (q1 – (1.5 x IQR)) of the number of mapped reads in samples passing the N-count threshold was calculated, and the maximum number of mapped reads in the six negative controls was recorded ("max neg control"). All samples were required to have a minimum of 1.5 x "max neg control" value of mapped reads, otherwise they were assigned "fail". If the "max neg control" value was above the $25^{th}$ percentile, the whole plate was assigned "fail" due to contamination. If the "max neg control" value was between the lower whisker point and $25^{th}$ percentile, samples were additionally filtered by their Ct value (from the RT-qPCR assay), where samples with Ct values above 30 or 32 were assigned "fail" (depending on contamination severity). The number of ambiguous base calls per genome was also used as an indicator of contamination. For the routine SARS-CoV-2 genome surveillance at SSI, samples with more than five ambiguous base calls are assigned "fail"; however, for the current study, the threshold was set at maximum 10 ambiguous base calls.

Genomes passing the quality check were analysed with Nextclade [20], using the standard "sars-cov-2" reference dataset which specifies changes relative to the Wuhan-Hu-1 reference genome. For the "Primer scheme" and "Reference genome" analyses (Table 1), Nextclade (version 2.14.0) was used with dataset version 2024-02-16—04-00-32z. For the "Ambiguous base calls over time" analysis (Table 1), Nextclade (version 3.5) was used with dataset version 2024-05-08--11–39-52Z.

## Analysis of mapped reads

Mapped read coverage was analysed for regions and positions of interest, by extracting coverage from the primer trimmed BAM files using samtools depth. For recurrent ambiguous base calls, the base calls and VAF-values were collected from the VCF files for the corresponding positions using a custom python script. All calculations and plots were done in RStudio (version 2023.3.0.386, RRID: SCR_000432) [21], with R packages tidyverse (RRID: SCR_019186) [22], gridExtra

(RRID: SCR_025249 [23], stringr (RRID: SCR_022813) [24], ggplot2 (RRID: SCR_014601) [25], and RcolorBrewer (RRID: SCR_016697) [26].

To investigate the mapping of individual reads in more detail, primer-trimmed BAM files of selected samples were loaded into the Integrative Genomics Viewer (IGV) (version 2.13.0, RRID: SCR_011793) [27], together with the relevant reference genomes and BED files. To further understand the differential mappings dependence on reference genome choice, a genome alignment of the three utilized reference genomes was generated with MAFFT (version 7.525, RRID: SCR_011811) [28], and visualized with JalView (version 2.11.4.1, RRID: SCR_006459) [29].

## Analysis of the impact of primer scheme on genome sequence

To determine consistency of base calls between samples when sequenced with different primer schemes, paired results were collected from the Nextclade output files for each sample using a custom python script. Only samples generating results with Nextclade for both members of the sample pair were analysed. For each paired sample, ambiguous base calls, substitutions, and undetermined positions were collected (columns: "nonACGTNs", "substitutions", "missing"), and compared within the pair. Positions with an undetermined base call in at least one sample were ignored. All other positions with an ambiguous base call in at least one sample were retained. Subsequently, a list of paired inconsistencies was generated and summarised for each primer scheme comparison.

## Analysis of ambiguous base calls following the emergence of the BA.2.86.x variant

To investigate whether ambiguous base calls become more prevalent with time, and in response to the emergence of a new variant, ambiguous base calls were collected for samples sequenced in the period where the BA.2.86.x variant took over. For this analysis, the ambiguity threshold was set to 0.8. All ambiguous base calls occurring in at least 5% of the filtered samples were selected for further analysis. For each of these recurrent ambiguous base calls, observed frequencies were calculated per week, with an additional filter of minimum 100 samples processed for sequencing per week. Likewise, for the same samples, the observed frequency of the BA.2.86.x variant was calculated per week.

## Removal of human reads from raw data

All raw data used in the current study has been submitted to ENA, after removal of unmapped reads (including human reads) with ReadItAndKeep [30]. To ensure that the filtered reads include data mapping to different reference genomes, a fasta-file containing both the Wuhan-Hu-1 reference strain genome and our currently implemented "BA2 consensus" reference genome (with removal of poly-A tail) was used as the "--ref_fasta" argument.

## Results

### Impact of primer scheme on base calls

The impact of primer scheme on the consensus sequences was investigated by sequencing 810 samples with two different primer schemes (as detailed in Table 1) and extracting base call information for all positions generating inconsistent results for the same sample. In total, 2207 and 395 inconsistent base calls were observed for each primer scheme comparison (V3 versus V4.1 and V3 versus V5.3.2), corresponding to an average of 3.8 and 1.7 inconsistent base calls per sample, respectively. Most of the inconsistencies were caused by an ambiguous base call observed with only one of the primer schemes. However, 74 and 27 inconsistencies corresponding to distinct base calls (without any ambiguity) were observed for each of the two primer scheme comparisons. Most inconsistencies were only observed once (82% and 90% of all observed inconsistencies per primer scheme comparison). However, some inconsistent base calls were found to be recurrent, indicative of non-random PCR artifacts or primer trimming issues. To gain further insights into the underlying causes, data for positions recurrently displaying inconsistent base calls was collected for further analysis.

A total of seven positions generated recurrent inconsistent base calls for more than 10% of the samples in the two primer scheme comparisons (Table 2). When samples were sequenced with Artic V3, ambiguous base calls were common at three positions (25,000, 26,529, 26,577), while generating substitutions for the same samples when sequenced with Artic V4.1 and V5.3.2 (Table 2). Notably, each of these positions are in primer regions for Artic V3 (primers 83L, 88L and 87R), with the minor base corresponding to the sequence of the primers (and the Wuhan-Hu-1 reference genome, on which they were designed). In all three cases, the dominant base of the ambiguous base calls matched the substitutions called with Artic V4.1 and V5.3.2 in over 98% of the samples. Interestingly, when inspecting the alignments of reads supporting the minor base in more detail (page 1 in S3 File), most were trimmed by the primer-trimming software, but a minor fraction remained untrimmed. These reads did not conform to the "rules" of the primer trimming software, either mapping in the opposite direction of the primer, or having alignments starting upstream of the primer region. Thus, if these reads indeed originate from the primers, they must represent chimeric sequences generated during PCR [10].

Recurrent ambiguous base calls were also observed at four positions in samples sequenced with Artic V4.1 (positions 8835, 14,960, 15,510, 15,521) (Table 2). However, unlike the ambiguous base calls generated with Artic V3, only one of the positions was in a primer binding site for Artic V4.1 (position 14,960), and no substitutions were observed with Artic V3 and V5.3.2 (Table 2). Notably, no variability was observed at these positions for any of the samples analysed, when sequencing with Artic V3 and V5.3.2 (S1-S2 Figs). Furthermore, the dominant base for the ambiguous base calls generated with Artic V4.1 was inconsistent with the base calls generated with Artic V3 and V5.3.2 in a large fraction of the samples (ranging from 14% of the samples at position 8835 to 52% of the samples at position 15,521). For position 15,521, nearly 10% of the samples had a substitution instead of an ambiguous base call (Table 2).

Since three of the positions with frequent ambiguous base calls in Artic V4.1 were in close proximity, the mapped read coverage was analysed in more detail for this region (Fig 1, page 2 in S3 File). Overall, amplicon 49 and 51 performed poorly compared to amplicon 50 and 52 for all samples with Artic V4.1. Notably, samples with ambiguous base calls only had partial coverage of amplicons 49 and 51 (Fig 1, highlighted in red), indicative of incorrect PCR amplification. Samples which did not generate ambiguous base calls with Artic V4.1 had coverage across amplicons 49 and 51 (Fig 1), suggesting that this PCR artifact is generally present but can be overpowered when the PCR amplification of the weakly performing amplicons is sufficiently strong.

**Table 2. Recurrent base calling inconsistencies for samples sequenced with Artic V3 and an alternative primer scheme.**

| Position | Artic V3 with spike-ins[a] | Artic V4.1[a] | Sample fraction Artic V3-V4.1[b] | Artic V5.3.2[a] | Sample fraction Artic V3-V5.3.2[b] |
|---|---|---|---|---|---|
| 8835 | T | C/T | 0.184 | – | 0 |
| 14,960 | A | A/T | 0.252 | – | 0 |
| | A | A→T | 0.005 | – | 0 |
| 15,510 | T | C/T | 0.248 | – | 0 |
| | T | T→C | 0.005 | – | 0 |
| 15,521 | T | A/T | 0.674 | – | 0 |
| | T | T→A | 0.095 | – | 0 |
| 25,000 | C/T | C→T | 0.350 | C→T | 0.057 |
| | C→T | – | 0 | C/T | 0.004 |
| 26,529 | A/G | G→A | 0.426 | G→A | 0.196 |
| | C/G | C/G | 0.002 | – | 0 |
| 26,577 | C/G | C→G | 0.469 | C→G | 0.217 |

a. Base-calls identical to the Wuhan-Hu-1 reference genome are shown as a single base letter, ambiguous base calls are shown with an upward slash (e.g. C/T), and substitutions are shown with an arrow (e.g. T→C). Inconsistencies not observed between Artic V3 and the alternative primer scheme are denoted with a dash.

b. The fraction of analysed sample pairs where the inconsistency was observed.

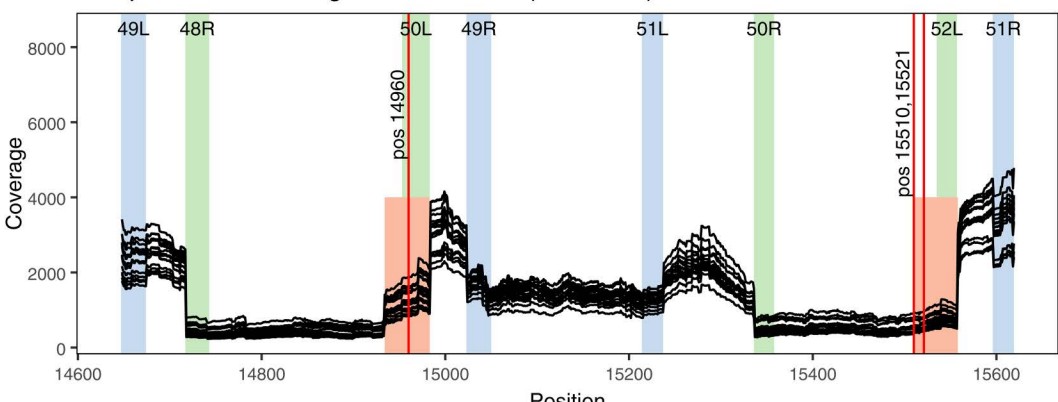

**Fig 1. Mapped read coverage for samples sequenced with Artic V4.1 in a region with frequent ambiguous base calls.** Per base coverage is shown as individual lines for each sample. Upper panel displays coverage for 17 samples which generated ambiguous base calls at positions 14,960, 15,510 and 15,521, while lower panel displays coverage for 17 samples which did not generate ambiguous base calls at these positions (accession numbers detailed in S2 Table). Positions on the x-axis correspond to the Wuhan-Hu-1 reference genome. The primer binding sites of Artic V4.1 are highlighted in blue for pool 1 and in green for pool 2, with amplicon number and orientation shown at the top of the highlighted region. The region with increased coverage for amplicon 49 and 51 is highlighted in red, with the three positions having frequent ambiguous base calls shown as vertical red lines.

No recurrent ambiguous base calls were observed with Artic V5.3.2, likely because the primer scheme is the most recent and presents a higher identity between the primers and samples analysed here.

Ambiguous base calls can also result from contamination. To investigate this possibility, we analysed the mapped read coverage across all control samples (144 negative controls in total, with six controls on each 96-well plate, as detailed in Methods section). Overall, control samples exhibited coverage below 10x for the large majority of genome positions (S4 File), with an average genome recovery of 0.2%. The regions with higher coverage were generally restricted to only one of the 6 negative controls on an individual sequencing plate, indicative of localized sample-to-sample contamination. Therefore, it is very likely that some of the base calling inconsistencies occurring at low frequency were the result of such contamination.

The genomic positions associated with recurrent inconsistent base calls listed in Table 2 showed depth coverage of zero in the majority negative controls, with a maximum coverage of 5x (S3 Table). For Artic V3, we found a total of 7 mapped reads across 72 negative controls at positions 25,000, 26,529 and 26,577, none of which supported the minor

bases responsible for the observed ambiguity. For Artic V4.1, there was no mapped read coverage at positions 8,835, 15,510 and 15,521 in any of the 48 negative controls used, while 38 reads mapped to position 14,960 (distributed across 15/48 controls). Thus, except for position 14,960, we found no evidence of the inconsistencies reported in Table 2 in any of the negative controls used with the primer schemes. Given that these base calling inconsistencies were observed with very high frequency across all the sequencing plates (S3-S4 Figs), we would expect to see the inconsistencies in at least some of the negative controls if they were the result of recurrent contamination.

### Impact of the reference genome on base calls

The quality of a reference-based genome assembly depends on the correct mapping of reads, which in turn depends on the (Hamming/genetic) distance between the reads and the reference. The BA.2.86.x variant [31–33] contains several new mutations, particularly in the spike-protein gene, which could challenge pipelines using the Wuhan-Hu-1 reference genome. We therefore tested the performance of our pipeline for 14 samples of the BA.2.86.x variant, using three different reference genomes (as detailed in Table 1). A region spanning positions 21,610 to 21,640 (beginning of the spike protein gene) was called as undetermined in 12/14 samples when reads from these samples were mapped to the Wuhan-Hu-1 reference genome, indicating that the reads might not be fully mapped (S4 Table). Indeed, a comparison of the coverage profiles when mapping against the Wuhan-Hu-1 versus the BA2 consensus genomes revealed a distinct dip in coverage specifically in this region (Fig 2). Inspection of read alignments confirmed that reads in this region were only partially mapped on the Wuhan-Hu-1 reference genome (page 3 in S3 File). However, outside this narrow region, the coverage profiles were nearly indistinguishable, and the total number of mapped reads was only increased with 261–3227 reads per sample (median number of mapped reads per sample 1,391,562 and 1,393,136, respectively, for the two reference genomes).

Still, the improved mapping uncovered several features that were hardly called at all (only in one to two samples) with the Wuhan-Hu-1 reference genome (S4 and S5 Tables). These new features included a 12 bp insertion (21608:TCATG-CCGCTGT), 3 SNVs (C21618T, C21622T, G21624C), and a 9 bp deletion (21633–21641). Interestingly, the Wuhan-Hu-1 and BA2 consensus reference genomes only differ by a single SNV and a 9 bp deletion in the region (page 5 in S3 File), which in this case is sufficient to have a major impact on read alignments (pages 3–4 in S3 File).

The BA2 consensus reference genome also gave rise to three new contiguous ambiguous base calls (Y:22032, M:22033, R:22034), observed in 6 to 7 out of 14 samples, which were only observed in a single sample with the Wuhan-Hu-1 reference genome and not at all with the JN.1.4 reference genome (S4-S6 Tables). Moreover, a 3 bp deletion in the spike-protein gene was observed with JN.1.4 at position 23,009–23,011, which was called as undetermined with the BA2 consensus reference.

In conclusion, multiple differences in genome sequences were found to be a consequence of reference genome choice, which had a direct impact on the read mapping (Fig 2, pages 3–4 in S3 File).

### Artic V5.3.2 generates ambiguous base calls with the BA.2.86.x variant

The Artic V5.3.2 primer scheme was developed in response to the emergence of the Omicron variant, and released in the beginning of 2023 [7]. Since then, the BA.2.86.x variant emerged in the middle of 2023 [33] and reached dominance by the end of the year. The BA.2.86.x variant contains multiple mutations relative to the previous Omicron variants [32,33], so new base calling inconsistencies may have emerged. To investigate whether this could be the case, we extracted all ambiguous base calls observed in samples prepared with the Artic V5.3.2 primer scheme in the period where variant BA.2.86.x took over (as detailed in Table 1) and plotted their prevalence over time (Fig 3).

A total of eight recurrent ambiguous base calls were identified in this data set (prevalence higher than 5% across 8865 samples, with an ambiguity threshold of 0.8) (Fig 3). Despite some weekly variation, the fraction of samples affected by ambiguous base calls shows an increasing trend, coincident with the takeover of the BA.2.86.x variant. Three of the

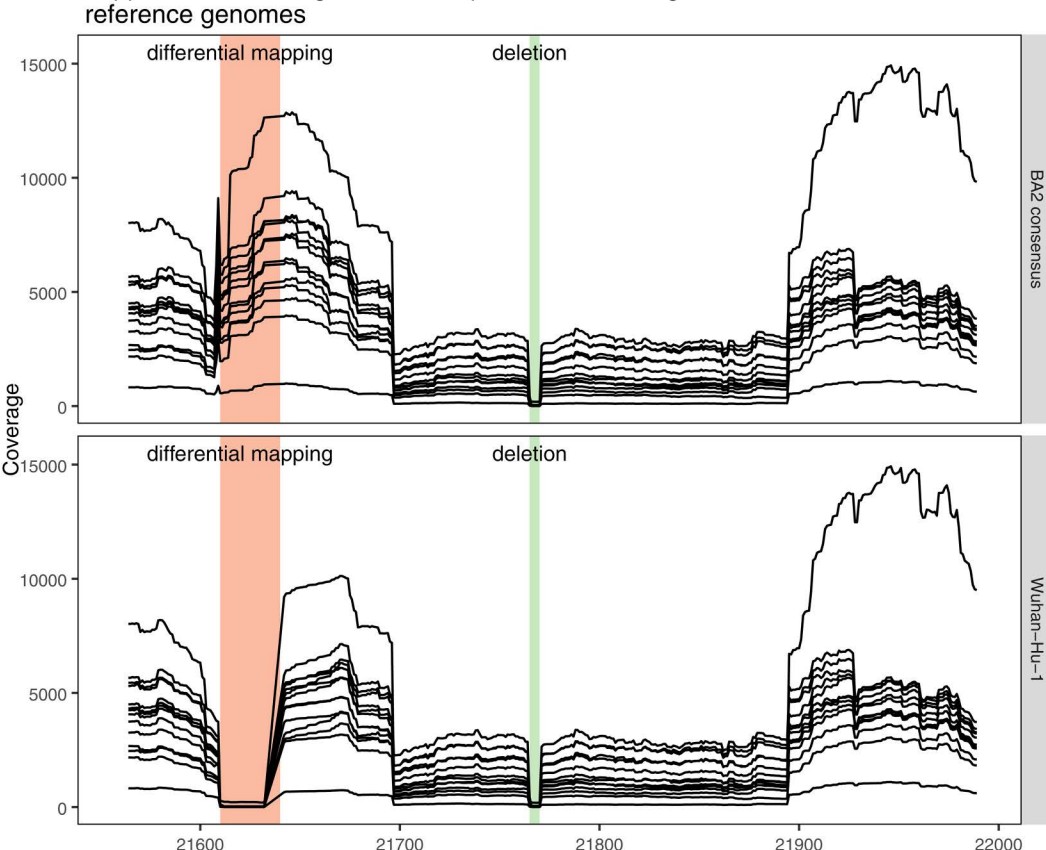

**Fig 2. Mapped read coverage for 14 samples sequenced with Artic V5.3.2, when mapped against two different reference genomes.** Per base coverage is shown as individual lines for each sample (accession numbers detailed in S2 Table). Upper and lower panels display coverage when mapping against the BA2 consensus and Wuhan-hu-1 reference genomes, respectively. Positions on the x-axis correspond to the Wuhan-Hu-1 reference genome. The region with differential mapping is highlighted in red. A deletion (relative to Wuhan-Hu-1) responsible for a second region without coverage is highlighted in green.

recurrent ambiguous base calls were at position 22,032−22,034, which were also noted in the "Reference genome" analysis. The larger dataset used here showed that 6% to 8% of the samples were affected by the misalignments in this region. For the other five recurrent ambiguous base calls, three were in primer binding sites (positions 13,427, 22,770 and 26,610). Out of these, two had substitutions relative to the Wuhan-Hu-1 reference (G22770A and A26610G) in other samples, indicating that they may be primer derived. However, without data from an alternative primer scheme (or samples sequenced without targeted PCR enrichment), it is difficult to evaluate the nature and severity of these possible new artifacts.

## Discussion

The current standard practice for sequencing SARS-CoV-2 genomes is based on targeted PCR enrichment, followed by shotgun sequencing and reference-based genome assembly. This approach has been cost-efficient for high-throughput surveillance during the pandemic, but it involves a significant investment in maintenance to ensure high genome quality.

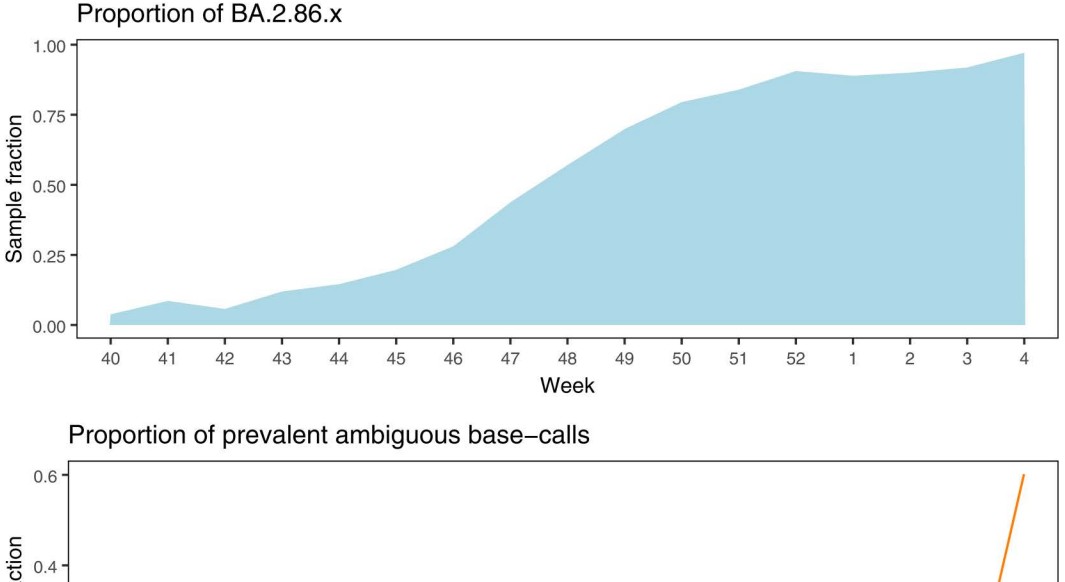

Proportion of BA.2.86.x

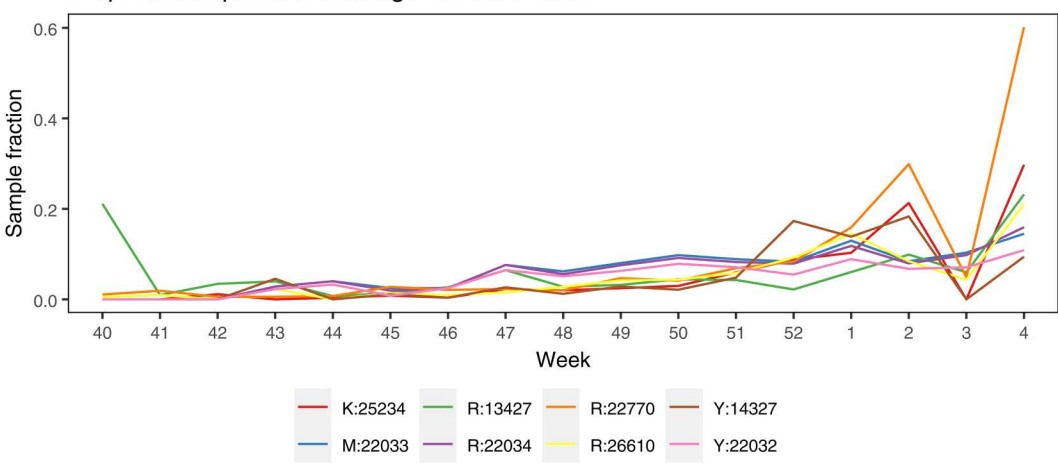

Proportion of prevalent ambiguous base–calls

Legend:
- K:25234
- R:13427
- R:22770
- Y:14327
- M:22033
- R:22034
- R:26610
- Y:22032

**Fig 3. Development in prevalence of frequent ambiguous base calls over time, during the take-over of the BA. 2.86.x variant.** The x-axis denotes week numbers from end of 2023 to beginning of 2024. Upper panel displays the proportion of samples classified as BA.2.86.x, while lower panel displays the proportion of each of 8 frequent ambiguous base calls for the same time period. The ambiguous base calls are named as per Nextclade, with the letter indicating the IUPAC ambiguity code and the number indicating the position relative to the Wuhan-Hu-1 reference genome.

The primer schemes developed by the Artic Network [6] for the Illumina platform consist of nearly 100 amplicons, each with a set of primers which are vulnerable to the evolution of the virus. Mutations in primer binding sites which do not result in a failed PCR reaction can also generate systematic errors in the assembled genomes if they are not handled correctly [34]. In the current study, we identified recurrent ambiguous base calls in multiple primer regions, despite having implemented primer trimming in our pipeline. While ambiguous base calls can also be the result of recurrent contamination, we found no evidence of such a contamination in our negative controls (72 negative controls, distributed on 12 sequencing plates). Inspection of the aligned reads did not uncover any issues with the primer trimming, as the reads generating ambiguity were mapped in the opposite direction of the primer or upstream of the primer region (page 1 in S3 File). Thus, if the reads originate from the primers, they are likely chimeric sequences formed by recombination during PCR [10]. Chimeras are known to arise from incomplete primer extension during PCR, where partially extended products act as primers on heterologous templates [10,35,36]. By having primers with bases distinct from the template in the PCR reaction, such a heterologous template is effectively generated.

Examples of read coverage inconsistent with the primer scheme used were observed with Artic V4.1, likewise indicative of incorrect PCR amplification. In this case, however, the artifact is likely the result of unintended primer

binding (a primer binding elsewhere than the target it was designed for), as has indeed been described previously for two of the positions [37,38]. In the current study, the PCR artifacts associated with Artic V4.1 were particularly problematic in case of amplicon drop-out (Fig 2), suggesting regions with amplicon dropout are vulnerable to such artifacts.

The reference genome was also found to significantly impact genome assembly. Notably, reads from samples of the BA.2.86.x variant were only partially mapped on the Wuhan-Hu-1 reference genome at the beginning of the spike-protein gene, resulting in multiple "defining mutations" [33] not being included in the genome assembly. Some of these issues could potentially be addressed by using mapping software with local realignment capability [11]. However, for the SARS-CoV-2 genome routine surveillance at SSI, we chose to replace the Wuhan-Hu-1 reference with a consensus genome of BA.2.x variants to avoid having to implement new software with our current pipeline. Moreover, as SARS-CoV-2 continues to evolve, inevitably, the reference will eventually need further replacement, regardless of mapping software. Considering the accumulated diversity of SARS-CoV-2 variants, a pipeline capable of using multiple reference genomes would be desirable to increase the likelihood of correctly mapping reads [39]. Alternatively, a pipeline implementing a combination of *de novo* assembly and contig alignment could potentially be more robust against artifacts originating from mis-aligned reads [34].

The need to continuously update both the primer scheme and pipeline is a concern, considering the global downscaling of SARS-CoV-2 genome sequencing. Since the emergence of new variants cannot be predicted, updates are, by necessity, done reactively, leaving a potentially significant period of incomplete genome sequencing, during which novel and epidemiologically relevant features may go unnoticed. As SARS-CoV-2 continues to evolve, it is also increasingly challenging to design universal primers [7], and backwards compatibility will likely be lost. Thus, it may be time to consider whether the targeted PCR enrichment strategy should be replaced or complemented by alternative approaches, such as direct metagenomic sequencing or others types of target enrichment [4]. Although more challenging, it needs to be weighed against the genome quality and cost of maintaining primer schemes and pipelines.

## Supporting information

**S1 Fig. Base calls generated for samples sequenced with Artic V3 and V4.1 at seven positions with frequent inconsistencies.** All base calls observed at each of the seven positions are shown on the x-axis, with the number of samples having the base call on the y-axis.
(PDF)

**S2 Fig. Base calls generated for samples sequenced with Artic V3 and V5.3.2 at seven positions with frequent inconsistencies.** All base calls observed at each of the seven positions are shown on the x-axis, with the number of samples having the base call on the y-axis.
(PDF)

**S3 Fig. Distribution of ambiguous base calls associated with Artic V3 across sequencing plates.** Each panel contains data for a plate of samples sequenced with two primer schemes, with the last six digits of the plate-id displaying the sequencing date. For each panel, the three positions where ambiguity is observed in Artic V3 with high frequency (see Table 2) are shown at the x-axis. The number of samples displaying inconsistency (ambiguous base call with Artic V3 and a clean base call with Artic V4.1/Artic V5.3.2) is shown on the y-axis.
(PDF)

**S4 Fig. Distribution of ambiguous base calls associated with Artic V4.1 across sequencing plates.** Each panel contains data for a plate of samples sequenced with both Artic V3 and Artic V4.1, with the last six digits of the plate-id displaying the sequencing date. For each panel, the four positions where ambiguity is observed in Artic V4.1 with high

frequency (see Table 2) are shown at the x-axis. The number of samples displaying inconsistency (ambiguous base call with Artic V4.1 and a clean base call with Artic V3) is shown on the y-axis.
(PDF)

**S1 Table. Spike-in primers developed and implemented at SSI for Artic V3.**
(XLSX)

**S2 Table. Accession numbers for samples plotted in** Fig 1 **and** Fig 2**.**
(CSV)

**S3 Table. Mapped read coverage in negative controls at positions with recurrent base calling inconsistencies in the primer scheme analysis.** Coverage is shown for all the positions with recurrent base calling inconsistencies (as listed in Table 2), for each sequencing plate, in each of the 6 negative controls used on the plate. In the "experiment" column, "exp1" and "exp2" refer to the analysis of primer schemes Artic V3 vs Artic V4.1 and Artic V3 vs Artic V5.3.2 respectively.
(TSV)

**S4 Table. Nextclade output file for 14 samples sequenced with Artic V5.3.2 and assembled with the Wuhan-Hu-1 reference genome.**
(XLSX)

**S5 Table. Nextclade output file for 14 samples sequenced with Artic V5.3.2 and assembled with the BA2 consensus reference genome.**
(XLSX)

**S6 Table. Nextclade output file for 14 samples sequenced with Artic V5.3.2 and assembled with the JN.** 1.4 reference genome.
(XLSX)

**S1 File. GISAID identifiers for genomes used to generate the BA2 consensus reference genome.**
(PDF)

**S2 File. Bash script with the commands used for processing reads to genomes.** The tools and commands are identical to the current pipeline used for SARS-CoV-2 routine genome sequencing at SSI.
(TXT)

**S3 File. IGV screenshots of mapped reads in primer trimmed BAM files.** Screenshot 1 (page 1) shows alignments of reads supporting the minor allele at position 26,577 (of putative primer-origin), which were left untrimmed. Screenshot 2 (page 2) shows alignment of reads supporting substitutions only observed with the Artic V4.1 primer scheme. Screenshot 3−4 (pages 3−4) shows how reads from a sample of the SARS-CoV-2 variant "BA.2.86.x" get partially mapped in a 30 bp region of the spike-protein gene against the Wuhan-Hu-1 reference genome, while getting aligned inside the region when mapping against our internal BA2 consensus reference genome. A screenshot of the alignment of the three reference genomes analyzed in the current study is shown on page 5, where the differences responsible for the differential mapping in the region can be inspected.
(PDF)

**S4 File. Mapped read coverage on negative controls for primer scheme analysis.** Mapped read coverage (depth) is shown for each of 24 sequencing plates used to generate the data in the "primer scheme" analysis (see Table 1). The titles of the plots denote the primer scheme used and experiment number ("exp1": Artic V3 vs Artic V4.1, "exp2": Artic V3 vs Artic V5.3.2). The plate-id for each sequencing plate consists of a plate-number (P1-P12, referring to a plate of samples

sequenced with two primer schemes), the sequencing date, and a run-number (plates sequencing on the same run have the same run-id). Each plate was sequencing with a total of 6 negative controls (see Methods section), for which coverage is plotted in the same panel. Dotted lines in the panels show the positions where ambiguous base calls were observed with high frequency for the corresponding primer scheme. The red horizontal line denote the 10x coverage. (PDF)

## Acknowledgments

The data in the current study is a tiny fraction of the more than one million SARS-CoV-2 samples sequenced as part of the routine surveillance done by Statens Serum Institut in Denmark since June 2021. This enormous sequencing effort was only made possible thanks to a large number of people who contributed to the data generation and logistics of sample processing in different ways. We would like to acknowledge the contribution of the laboratory technicians at the Department of Sequencing and Bioinformatics (Emine Yüksel Coskun, Anna Rønberg Shmidt-Nielsen, Yonos Hariesi, Kirsten Henneberg), the Department for Congenital disorders (Jacob Sønderby Pedersen, Malihe Nikou-Vedadradi, Arzu Caglar Nemli, Helene Kjerulf), and TestCenter Denmark (Sofie Skov Petersen, Mohammad El-Najjar, Morten Warring). We would also like to acknowledge the team at the section of System Development & Data Integration, at the Department of Digital Infrastructure, for their work on automating laboratory procedures and data flow.

## Author contributions

**Conceptualization:** Kirsten Maren Ellegaard.

**Data curation:** Kirsten Maren Ellegaard, Vithiagaran Gunalan, Raphael Sieber, Marc Bennedbæk, Leandro Andres Escobar-Herrera, Gitte Nygaard Aasbjerg, Casper Westergaard.

**Formal analysis:** Kirsten Maren Ellegaard.

**Investigation:** Kirsten Maren Ellegaard, Vithiagaran Gunalan, Sharmin Jamshid Baig, Nicolai Balle Larsen, Jonas Bybjerg-Grauholm, Tobias Gress, Theis Hass Thorsen, Anders Krusager, Nour Saad Al-Tamimi, Christina Wiid Svarrer.

**Software:** Kirsten Maren Ellegaard, Vithiagaran Gunalan, Raphael Sieber, Marc Bennedbæk, Leandro Andres Escobar-Herrera, Gitte Nygaard Aasbjerg, Casper Westergaard.

**Validation:** Kirsten Maren Ellegaard, Vithiagaran Gunalan, Raphael Sieber, Sharmin Jamshid Baig, Nicolai Balle Larsen, Marc Bennedbæk, Jonas Bybjerg-Grauholm, Leandro Andres Escobar-Herrera, Tobias Gress, Theis Hass Thorsen, Anders Krusager, Gitte Nygaard Aasbjerg, Casper Westergaard, Christina Wiid Svarrer.

**Visualization:** Kirsten Maren Ellegaard.

**Writing – original draft:** Kirsten Maren Ellegaard, Marc Stegger.

**Writing – review & editing:** Kirsten Maren Ellegaard, Vithiagaran Gunalan, Raphael Sieber, Sharmin Jamshid Baig, Nicolai Balle Larsen, Marc Bennedbæk, Jonas Bybjerg-Grauholm, Leandro Andres Escobar-Herrera, Tobias Gress, Theis Hass Thorsen, Anders Krusager, Gitte Nygaard Aasbjerg, Nour Saad Al-Tamimi, Casper Westergaard, Christina Wiid Svarrer, Morten Rasmussen, Marc Stegger.

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
