## [Decision Letter · Decision Letter 0]

26 May 2025

Dear Dr. Ellegaard,

Thank you for submitting your manuscript to PLOS ONE. After careful consideration, we feel that it has merit but does not fully meet PLOS ONE’s publication criteria as it currently stands. Therefore, we invite you to submit a revised version of the manuscript that addresses the points raised during the review process.

We look forward to receiving your revised manuscript.

Kind regards,

Nihad A.M Al-Rashedi

Academic Editor

PLOS ONE

Journal Requirements:

2.  Please note that PLOS ONE has specific guidelines on code sharing for submissions in which author-generated code underpins the findings in the manuscript. In these cases, we expect all author-generated code to be made available without restrictions upon publication of the work. 

Please review our guidelines at https://journals.plos.org/plosone/s/materials-and-software-sharing#loc-sharing-code and ensure that your code is shared in a way that follows best practice and facilitates reproducibility and reuse.

3. We are unable to open your Supporting Information file [Table_S3 to Table_S5(.tsv)]. Please kindly revise as necessary and re-upload.

4. We note that there is identifying data in the Supporting Information file <File_S3.csv>. Due to the inclusion of these potentially identifying data, we have removed this file from your file inventory. Prior to sharing human research participant data, authors should consult with an ethics committee to ensure data are shared in accordance with participant consent and all applicable local laws.

-Location data

Please remove or anonymize all personal information, ensure that the data shared are in accordance with participant consent, and re-upload a fully anonymized data set. Please note that spreadsheet columns with personal information must be removed and not hidden as all hidden columns will appear in the published file.

**Additional Editor Comments:**

Your publication has been analyzed by ScreenIT, an independent group of scientists dedicated to developing automated tools for the assessment of academic papers. Our analysis offers insights into various aspects of your research, such as originality, clarity, and overall contribution to the field. The screening tools are designed to support efforts in enhancing and maintaining high-quality standards. Please find the detailed report attached in the PDF.

Reviewers' comments:

Reviewer's Responses to Questions

**Comments to the Author**

1. Is the manuscript technically sound, and do the data support the conclusions?

Reviewer #1: Yes

Reviewer #2: Partly

2. Has the statistical analysis been performed appropriately and rigorously?

Reviewer #1: Yes

Reviewer #2: N/A

3. Have the authors made all data underlying the findings in their manuscript fully available?

Reviewer #1: Yes

Reviewer #2: Yes

4. Is the manuscript presented in an intelligible fashion and written in standard English?

Reviewer #1: Yes

Reviewer #2: Yes

Reviewer #1: This is an automated report for PONE-D-24-44224. This report was solicited by the PLOS One editorial team and provided by ScreenIT.

ScreenIT is an independent group of scientists developing automated tools that analyze academic papers. A set of automated tools screened your submitted manuscript and provided the report below. Each tool was created by your academic colleagues with the goal of helping authors. The tools look for factors that are important for transparency, rigor and reproducibility, and we hope that the report might help you to improve reporting in your manuscript. Within the report you will find links to more information about the items that the tools check. These links include helpful papers, websites, or videos that explain why the item is important. While our screening tools aim to improve and maintain quality standards they may, on occasion, miss nuances specific to your study type or flag something incorrectly. Each tool has limitations that are described on the ScreenIT website. The tools screen the main file for the paper; they are not able to screen supplements stored in separate files. Please note that the Academic Editor had access to these comments while making a decision on your manuscript. The Academic Editor may ask that issues flagged in this report be addressed. If you would like to provide feedback on the ScreenIT tool, please email the team at ScreenIt@bih-charite.de. If you have questions or concerns about the review process, please contact the PLOS One office at plosone@plos.org.

Reviewer #2: This article deals with some important questions concerning viral genome sequencing and in general the article is well written and coherent. The article focuses on how the choice of tools which are used in that process, both lab reagents and bioinformatic software, can impact the resulting consensus genomes. Due to the widespread adoption of these techniques, it is important to investigate these potential issues, and I commend the authors for highlighting concerns around primer choices. I also commend the authors for making all of their sequences available, as well as the code used in consensus sequence generation.

There have been plenty of examples of amplicon-based primer schemes producing spurious results due to unexpected primer binding, as well as the use of inappropriate bioinformatics tools which can lead to primers not being correctly removed from the resulting data, and this paper seeks to add to that knowledge base. Unfortunately, at the moment I do not believe that the authors have conclusively proven some of the points which they have set out to make, for reasons which I will outline below, but with some further work they should be able to do so.

Major Issues:

Firstly, the negative contamination criteria seems problematic to me in the context of this study (detailed in lines 217 to 232). From my understanding of it, the negative controls can have any number of mapped reads up to the 25th percentile, in which case some samples will be classed as a failure depending on their ct value. This seems overly permissive of contamination to me. With a median mapped read value of around 1.4 million for some samples (line 360), you could imagine a scenario where the negative control could have thousands of mapped reads and the whole run would not be considered contaminated. Furthermore, due to the variable sequencing depth achieved with amplicon methods, as the authors themselves have outlined in line ~436 of the discussion and line 328, the incorporation of unexpected amplicons can easily lead to spurious calls in regions of low coverage in a particular sample. This is further highlighted by the authors when they state that they increased their cut-off for ambiguous bases from 5 to 10 for this study (line 232), a metric they normally use to indicate contamination of the run.

It is possible that contamination is not an issue with these samples, but the way the negative control filtering has been described and performed means that as a reader I feel I don’t have enough information to rule out the possibility that the results are due to contamination. This in turn diminishes the credibility for the findings in the “Impact of primer scheme on base calls” section. In my opinion almost all of the claims in this section could be due to contamination, especially as ambiguous base calls are a hallmark of run contamination.

To make the claims in this section sound, it is my opinion that the authors need a much stricter criteria for excluding samples based on potential negative contamination. A frequently used cut-off would be if there is more read depth for a given site in the negative control than the depth of the consensus caller (10 reads in this study (line 213)), the run would be considered contaminated and would not be taken forwards. The authors at least need to detail how many reads mapped and what the maximum depth of their negative controls were in order to satisfactorily rule out contamination as being the root cause of these mixed bases.

Secondly, the authors frequently suggest that mis-mapping of reads is the cause of some of the artefacts seen throughout the paper but don’t demonstrate this in any way. Mis-mapping reads can be seen if the bam files are inspected, or perhaps there would be an increase or decrease in the number of reads mapped at a given site. For example, lines 377-379, the authors claim that there is a difference in how the reads map in this region but show no evidence of this, and it could be easily viewed in (e.g.) IGV. In lines 381-384, the authors say that there is better mapping in this region but how is this measured?

It seems to me that there is a reasonable chance that a proportion of these findings are issues with the bioinformatics tools used in the genome assembly, rather than an inherent primer scheme issue. For example, are the positions listed in line 297 all in primer binding sites? If ambiguous calls are in primer sites is that an indication of incomplete primer trimming by iVar? The authors allude to the primer trimming software being the issue in line 433. It feels to me that this could be investigated further and could be the root cause of some of these artefacts rather than the primer scheme. It is a worthwhile finding to point out flaws in commonly used bioinformatics tools, but this is not investigated currently.

Thirdly, with regards to figure 3, it seems to me that this graph ends too early. There is some increase in the ambiguous calls over time certainly, but the lineage prevalence is essentially the same for weeks 52,1,2,3,4, whereas the fraction of ambiguous bases doesn't really change until week 2, then dips, then is up for week 4. How do weeks 5+ look? If your rational is correct it will presumably continue increasing or plateau at this higher level? I think this figure needs to be expanded.

Minor issues:

Line 45: That reference appears to not say that the virus is becoming less virulent – “A popular and incorrect view on the evolution of virulence, frequently expressed in the context of SARS-CoV-2, is that in the long run, pathogens will tend to evolve to be decreasingly virulent” (https://www.nature.com/articles/s41579-023-00878-2)

line 294: How many were left after this criteria?

line 307: I don't understand how there is a range of % here.

line 325: Can the authors investigate what mechanism would generate this artefact? Could a different primer be binding to that region? Or is the library prep fragmentation process favouring that area?

line 328: This is the exact scenario where a permissive definition of contamination in the negative control can lead to mixed/erroneous base calls.

line 395: Samples sequenced in the author’s lab?

line 430: Where the snp falls in a given read can be investigated by looking at the alignments and would go some way towards answering this question.

**Do you want your identity to be public for this peer review?** For information about this choice, including consent withdrawal, please see our Privacy Policy

Reviewer #1: No

Reviewer #2: No

---

## [Author Response · Author response to Decision Letter 1]

7 Jul 2025

We would like to thank the editor and reviewer for the swift and constructive feedback on the manuscript!

For the major points of the revision, we have added new data and analysis on the negative controls used for the Primer scheme analysis, which we believe should address the possible role of contamination for this part of the study. We have also added multiple screenshots from IGV, where read alignments in bam-files can be inspected in more detail for regions of particular interest. All specific comments from both editor and reviewers have been addressed in the submitted file "Response to Reviewers".

We believe the manuscript has benefited greatly from the revision, and hope that both editor and reviewer agree.

---

## [Decision Letter · Decision Letter 1]

4 Aug 2025

Dear Dr. Ellegaard,

We look forward to receiving your revised manuscript.

Kind regards,

Nihad A.M Al-Rashedi

Academic Editor

PLOS ONE

Journal Requirements:

Reviewers' comments:

Reviewer's Responses to Questions

**Comments to the Author**

Reviewer #2: (No Response)

2. Is the manuscript technically sound, and do the data support the conclusions?

Reviewer #2: Partly

3. Has the statistical analysis been performed appropriately and rigorously?

Reviewer #2: N/A

4. Have the authors made all data underlying the findings in their manuscript fully available?

Reviewer #2: Yes

5. Is the manuscript presented in an intelligible fashion and written in standard English?

Reviewer #2: Yes

Reviewer #2: Response to Authors’ responses

I thank the authors for the efforts they have made towards resolving the concerns I highlighted in the first round of review. I think that the work has made positive steps forward and a lot of my issues have been resolved. However, I still have concerns regarding the base calling inconsistencies that the authors have detailed.

I disagree that the contamination shown is only minor. Having positions with over 9000x coverage clearly shows that you have significant levels of negative control contamination (exp2: articV5). Your choice of dots for the plot have made it difficult to judge the exact size of the areas of contamination but presumably they are in the region of a 100 bases+. I also disagree with the assertion that “it is unlikely that contamination by chance should produce recurrent ambiguous base calls (affecting more than 10% of the samples) in a primer-scheme-specific manner.” Recurrent low-level ambiguous base calls would be exactly the sort of thing that would be visible when contamination occurs across a plate. I reject the assertion that this was a necessary sacrifice for surveillance sequencing purposes. For example the COG-UK consortium sequencing teams worked to a standard where if there was more than 100 mapped reads in a negative, the run would be considered contaminated (https://www.medrxiv.org/content/10.1101/2021.10.09.21264695v1.full;
https://pmc.ncbi.nlm.nih.gov/articles/PMC8461472/).

More importantly though, as the authors themselves have stated these SNPs are all in primer binding sites so this is really an issue with primer trimming software. I agree that the SNPs are potentially caused by poor primer trimming, but then that isn’t a primer scheme issue per se it is a bioinformatics issue and as such should be an issue on the iVar github page. However, the SNP at position 26577 is on the leading strand at that site (as shown in S3 file) and is due to not having a read be trimmed. There shouldn’t be an amplicon produced as there isn’t a forward primer expected to bind there so a trimming sorftware would not look at those reads? Can the authors suggest beyond that it is a primer trimming artefact what is going on for these sites? What is it about V4.1 that is causing this?

If these SNPs are real PCR artefacts beyond an issue in the lab, the authors should be able to show that these SNP artefacts can be seen across the enormous selection of SARS-CoV-2 sequences sequenced with the ARTIC primers that are out there. Clearly the SNPs that others have identified can be assumed to be larger issues, but what about the SNPs that are new to this study, can they be seen in samples further afield? Presumably they would appear as low frequency variants elsewhere too.

In summary, I think the authors need to demonstrate that these SNPs are an issue that affects others outside of their lab by showing they are present in SARS-CoV-2 genomes on a public repository, given that their sequencing run shows signs of contamination.

**Do you want your identity to be public for this peer review?** For information about this choice, including consent withdrawal, please see our Privacy Policy

Reviewer #2: No

---

## [Author Response · Author response to Decision Letter 2]

1 Sep 2025

We appreciate the time and effort the reviewer has put into the review, and hereby submit our rebuttal addressing the remaining concerns.

With the current revised manuscript, we have added a lot more analyses and details on the distribution of contamination. We think that the manuscript is now completely transparent in regard to contamination, the report on the extent and distribution of contamination is very detailed. We hope that these details have put the reviewer at ease regarding the contamination observed and its possible impact on the results we report.

---

## [Decision Letter · Decision Letter 2]

23 Sep 2025

SARS-CoV-2 sequencing artifacts associated with targeted PCR enrichment and read mapping

PONE-D-24-44224R2

Dear Dr. Ellegaard,

We’re pleased to inform you that your manuscript has been judged scientifically suitable for publication and will be formally accepted for publication once it meets all outstanding technical requirements.

Kind regards,

Nihad A.M Al-Rashedi

Academic Editor

PLOS ONE

Additional Editor Comments (optional):

Reviewer #2:

Reviewers' comments:

Reviewer's Responses to Questions

**Comments to the Author**

Reviewer #2: All comments have been addressed

2. Is the manuscript technically sound, and do the data support the conclusions?

Reviewer #2: Yes

3. Has the statistical analysis been performed appropriately and rigorously?

Reviewer #2: N/A

4. Have the authors made all data underlying the findings in their manuscript fully available?

Reviewer #2: Yes

5. Is the manuscript presented in an intelligible fashion and written in standard English?

Reviewer #2: Yes

Reviewer #2: Thank you to the authors for addressing my concerns.

**Do you want your identity to be public for this peer review?** For information about this choice, including consent withdrawal, please see our Privacy Policy

Reviewer #2: No

---

## [Editor Report · Acceptance letter]

PONE-D-24-44224R2

PLOS ONE

Dear Dr. Ellegaard,

I'm pleased to inform you that your manuscript has been deemed suitable for publication in PLOS ONE. Congratulations! Your manuscript is now being handed over to our production team.

Kind regards,

on behalf of

Dr. Nihad A.M Al-Rashedi

Academic Editor

PLOS ONE